# Investigating the application of IoT mobile app and healthcare services for diabetic elderly: A systematic review

**Jinglong Li**[1], **Rosalam Che Me**[1,2*], **Faisul Arif Ahmad**[3], **Qisen Zhu**[4]

**1** Department of Industrial Design, Faculty of Design and Architecture, Universiti Putra Malaysia, Serdang, Malaysia, **2** Malaysian Research Institue on Ageing (MyAgeing), Universiti Putra Malaysia, Serdang, Malaysia, **3** Department of Computer and Communication Systems Engineering, Faculty of Engineering, Universiti Putra Malaysia, Seri Kembangan, Selangor, Malaysia, **4** Faculty of Education, Universiti Kebangsaan Malaysia, Bangi, Selangor, Malaysia

* rosalam@upm.edu.my

## Abstract

As the prevalence of diabetes increases among the elderly population, effective management becomes increasingly crucial. IoT mobile applications offer promising solutions for diabetes care by providing real-time monitoring, medication management, and lifestyle support. This paper aims to investigate the potential applications and challenges of IoT mobile applications in managing diabetes among elderly patients. Three databases including Scopus, Web of Science, and IEEE were systematically searched; 29 articles were screened in the final analysis process. Key results indicate that the application of mobile apps includes blood glucose monitoring, medication adherence, promotion of physical activity, and dietary control. Devices such as continuous glucose monitors and smart pill dispensers significantly improve glycemic control and medication adherence rates, these technologies enable real-time tracking, personalized feedback, and timely interventions, which enhance self-management and communication with healthcare providers. However, technical challenges like interoperability, data security and privacy, usability and involvement of policymakers pose significant barriers to their effective implementation. Collaborative efforts from healthcare providers, device manufacturers, and policymakers are essential to overcome these barriers and fully leverage the benefits of IoT technologies in diabetic elderly care. This review highlights the need for collaborative efforts to develop standardized frameworks that ensure device compatibility and seamless data integration in IoT solutions for diabetic elderly care, enhance data privacy with advanced technology, and design user-friendly apps for the diabetic elderly to improve the generalization and adoption of IoT IoT mobile applications in healthcare fields for elderly.

## Introduction

Over the decades, the growing number of diabetes has become a public health concern, particularly among the elderly population due to their high development of diabetes [1]. The increasing prevalence of diabetes in the ageing population is caused by several factors such as

**Data availability statement:** All relevant data are within the manuscript and its Supporting Information files.

**Funding:** This research was funded by a grant from Universiti Putra Malaysia Geran Putra grant (GP-IPS). NO: [GP-IPS/2023/9772400]. The funders had no role in study design, data collection and analysis, decision to publish, or preparation of the manuscript.

**Competing interests:** NO authors have competing interests.

ageing, sedentary lifestyles, and unregular dietary habits [2]. Managing diabetes in the elderly population presents huge challenges, including multiple complications, age-related physiological changes, and the need for continuous healthcare services to prevent complications [3]. Hence, diabetic elderly patients suffer serious issues in their daily lives such as medication adherence, regular blood glucose monitoring, and maintaining a healthy lifestyle [4]. Cognitive decline and physical limitations among diabetic elderly would further complicate self-management. Therefore, special healthcare for the diabetic elderly is urgent and important [5].

In recent years, the development of Internet of Things (IoT) technology has experienced a soar revolution and developed into various sectors, including healthcare [6]. IoT mobile applications, equipped with sensors and connected devices, can offer innovative solutions to improve chronic disease management [7]. These applications facilitate real-time data collection, remote monitoring, and personalized feedback, making them particularly suitable for managing conditions like diabetes in the elderly population [8]. IoT mobile applications enable to solution those of challenges faced by elderly diabetic patients. For instance, IoT-enabled glucometers and continuous glucose monitors (CGMs) can provide real-time tracking of blood glucose levels, alerting patients and healthcare providers to any abnormal readings. Medication management apps can remind patients to take their medications and track adherence [9]. Fitness trackers and dietary monitoring apps can help patients maintain a healthy lifestyle by providing personalized recommendations and feedback [10].

Despite the potential, IoT applications for diabetic elderly management still face several issues nowadays [11]. Technical challenges such as interoperability between devices, data security and privacy concerns, and the need for reliable wireless connectivity are significant barriers [12]. Additionally, diabetic elderly often find it difficult to use and accept these new mobile applications due to their cognitive and physical limitations and declines, necessitating user-friendly designs and comprehensive training and support [13].

The purpose of this systematic review paper is to investigate the current and development status of IoT mobile applications and healthcare services designed for diabetic elderly patients and to explore the benefits, challenges, and potential future directions of these technologies, emphasizing their impact on patient outcomes and quality of life. By identifying and addressing the current issues and status, this paper seeks to highlight how IoT applications can be optimized to better provide healthcare to serve the needs of the diabetic elderly as well as an insight review result for researchers.

## Methodology

### Search strategy

This research adopted a systematic review of related articles that focus on a specific topic and organizes a review framework rather than a sample of (mainstream) journals and conferences during a limited time: the IoT mobile applications for diabetic elderly in the healthcare service context [14]. A systematic review was searched through those databases of Scopus, Web of Science (WoS), and IEEE Xplore (IEEE), and related articles were searched according to keywords in this search strategy (Terwee et al., 2009). Adopting the person, exposure (intervention), comparison, outcome (PECO/ PICO/ PIO) format search approach [15], articles were searched through the search terms: P: "diabet* old*", "diabet* elder*", "diabet* aged", "diabet* senior*"; I: "mobile app*", "IoT", "application*", "mobile", "app*", "smart technolog*", "digital solution*"; O: "healthcare", "health care", "service*", "chronic disease management*".

Through the PIO search terms, keywords (("diabete* old*" OR "diabete* elder*" OR "diabete* aged" OR "diabete* senior*") AND ("mobile app*" OR IoT OR application * OR mobile OR app*) AND (healthcare OR "health care" OR service*)) are keyed in to search for specific research in databases (Scopus, WoS, and IEEE). The search period time was set to cover the publications from 2014 to 2024. As a search result listed in Table 1, a total of 147 articles were found.

## Study selection

In the process of selecting the articles, the 147 articles will be carefully selected and removed according to the criteria listed in Table 2. Before proceeding to the next screening process, duplicate articles were removed. For those articles that go through the screening process, the full-text article of each study was independently examined by the author.

## Assessment of Bias

Although the purpose of this systematic review is descriptive, we assessed the risk of bias using the Cochrane risk-of-bias tool for randomized trials version 1 (RoB 1) [17]. Cochrane risk-of-bias tool is a commonly used tool to assess studies. The tool evaluates six domains, namely: random sequence generation, allocation concealment, blinding of participants and personnel, blind of outcomes assessment, incomplete outcome data and selective reporting. –,? and + represents either a high, unknown or low risk of bias, respectively. Within each domain, assessments are made for one or more items, which may cover different aspects of the domain, or different outcomes. If any domain was not rated "low", the overall risk of bias was considered "high". The assessment is attached to S2 File, (Sheet: Cochrance RoB1) includes the risk of bias assessment.

**Table 1. Search strings from Scopus, WoS and IEEE.**

| SCOPUS | TITLE-ABS-KEY ("diabete* old*" OR "diabete* elder*" OR "diabete* aged" OR "diabete* senior*") AND TITLE-ABS-KEY ("mobile app*" OR IoT OR application * OR mobile OR app* OR "smart technolog*" OR "digital solution*") AND TITLE-ABS-KEY (healthcare OR "health care" OR service * OR "chronic disease management*") | 79 results |
|---|---|---|
| WoS | (TS = ("diabete* old*" OR "diabete* elder*" OR "diabete* aged" OR "diabete* senior*") AND TS = ("mobile app*" OR IoT OR application * OR mobile OR app* OR "smart technolog*" OR "digital solution*")AND TS = (healthcare OR "health care" OR service * OR "chronic disease management*")) | 61 results |
| IEEE | ("diabete* old*" OR "diabete* elder*" OR "diabete* aged" OR "diabete* senior*") AND ("mobile app*" OR IoT OR application * OR mobile OR app* OR "smart technolog*" OR "digital solution*") AND (healthcare OR "health care" OR service * OR "chronic disease management*") | 7 results |

**Table 2. Inclusion and exclusion criteria [16].**

| Inclusion Criteria | Exclusion Criteria |
|---|---|
| (a) Articles include healthcare mobile applications as technology for diabetic elderly; (b) Articles focus on diabetic elderly's healthcare; (c) Access to the full articles. | (a)IoT mobile application but not healthcare application for diabetic elderly; (b)Mobile application does not focus on healthcare; (c)Healthcare mobile application design not for diabetic elderly; (d)Inaccessibility to full-text articles. |

## Assessment of study quality

Each article's quality was assessed by the Critical Appraisal Skills Programme (CASP)'s Quality Appraisal Tool systematic review checklist, and the result is presented in the form of a table. CASP is a tool that assists researchers in critically evaluating the quality of research studies. It provides a structured framework for assessing various aspects of study design, methodology, analysis, and reporting, helping users determine the trustworthiness, relevance, and validity of research findings [18]. The following part will emphasize the application and usage of CASP to evaluate each article in the screen section.

## Results

### Screen the article results

The PRISMA screen flow is presented in Fig 1. Through a systematic search, a total of 147 articles were identified, later 33 of those articles were found to be duplicates and were eliminated. As a result, 114 articles still needed to be screened. The first screening process was the title screening, unrelated title articles were removed in this process, accounting for 56 articles; abstracts that did not focus on topics were also removed. Therefore, a final 10 articles were removed. The remaining articles were further screened according to inclusion and exclusion criteria; therefore, another 19 articles were removed. The remaining 29 articles were assessed for eligibility and extracted for literature review. Table 3 recorded the quality evaluation results of the CASP appraisal for all included articles, the quality of those total 29 articles was

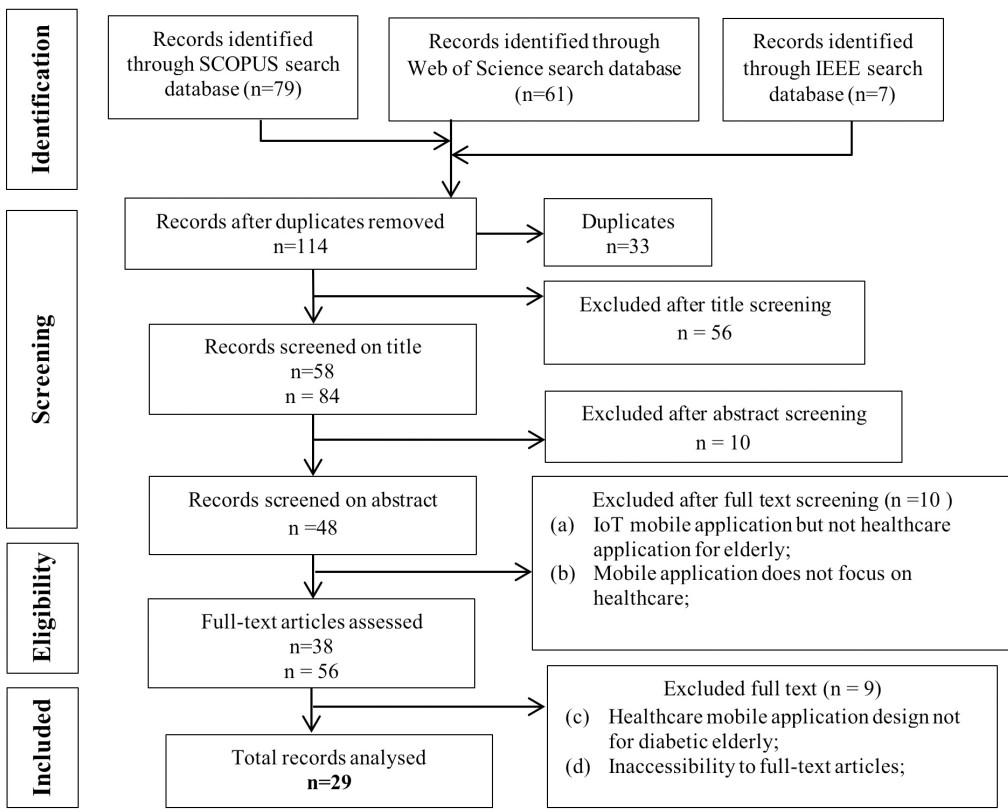

**Fig 1. Flow chart of search strategy based on PRISMA flow diagram.**

systematically assessed using the CASP checklist. The CASP tool focuses on key aspects such as study aims, recruitment methods, exposure and outcome measurement, and the handling of confounding factors. In this table, 29 articles are rated using a color-coded system: green indicates the criterion was met, orange suggests uncertainty or incomplete information, and red signifies the criterion was not fulfilled. Most articles demonstrated strong methodological rigor, with green markings across most evaluation criteria. However, a few articles were found lacking in addressing confounding factors, as seen by red or orange marks in the "Identified confounding?" and "Addressed confounding?" columns. Despite some variation in addressing confounding factors, all 29 papers ultimately met the necessary qualifications for inclusion, as they demonstrated sufficient methodological rigor and relevance across the key CASP criteria. This evaluation highlights the overall quality of the research while identifying specific areas of methodological concern, providing a clear basis for assessing the reliability and applicability of the findings in the context of a systematic review.

## Data extraction and analysis

Several attributes were coded and analysed under this study's research aims. These characteristics are as follows: The remaining research articles were subjected to quality screening based on the various inclusion and exclusion criteria mentioned in Table 2. Therefore, the final articles were reviewed down to 29 articles as shown in Fig 1. These articles were incorporated into the final step of data coding and analysis. According to the final screened articles, three themes can be classified, they were: (a) Health problems among diabetic elderly; (b) The application of IoT mobile app for diabetic elderly healthcare services; (c) The challenges of IoT mobile app for diabetic elderly healthcare services.

Missing data can arise at various stages in a systematic review, including missing studies, outcomes, or summary data, and addressing these issues is crucial for ensuring the reliability of the findings. To minimize the risk of missing studies, we conducted an extensive literature search across multiple databases. Additionally, contacting authors directly was employed as a strategy to retrieve unpublished work or additional data. For missing summary data, methods such as extracting data from figures, using available online tools, or calculating results from raw data were applied when feasible. These approaches helped mitigate the impact of missing data and ensured a more comprehensive analysis.

## Health problems among diabetic elderly

7 studies focused on talking about the health problems among diabetic elderly. The synthesis and details of these 7 studies are presented in Table 4, including publication year, thematic focus, research objectives, geographical context, and methodological data particulars in each article.

As a widespread chronic disease among the elderly population, diabetes is often neglected by the elderly population in their daily care. Diabetic elderly are facing health problems, against which they have specific characteristics and demands [27]. According to the screened articles, four serious and common diabetic-related issues are concluded suffered by diabetic elderly, including complications, hypo glycaemia, frailty and disability, and cognitive impairment.

Firstly, complications are common issues popular in the diabetic elderly. They are at higher risk of suffering microvascular and macrovascular diseases compared to younger people [39]. Complications will have a serious influence on diabetic elderly's living independence, self-care capability and daily life quality. Especially the episodes of cardiovascular and hypoglycemics are at the highest occurrences. According to the International Diabetes Federation worldwide

**Table 3. Quality assessment results of each included article using the CASP tool.**

| | A. Ait-Younes; F. Blanchard; B. Delemer; M. Herbin, 2014 [19] | Alan Sinclair, Trisha Dunning, Leocadio Rodriguez-Mañas, 2015 [20] | Fatemeh Mehravar et al., 2016 [21] | Singh, K, et al., 2016 [22] | Sircar, M, et al., 2016 [23] | Turki Alanzi, et al., 2018 [24] | Insa Feinkohl, et al., 2019 [25] | A. Matthew Prina, et al., 2019 [26] | Ni Putu W. P. S, et al., 2019 [27] | Jing Li, et al., 2020 [28] | Alberto Pilottoa, et al., 2020 [29] | Elena Toschi, et al., 2020 [30] | Arriel Benis, et al., 2020 [31] |
|---|---|---|---|---|---|---|---|---|---|---|---|---|---|
| Valid aim? | green | green | green | green | green | green | green | green | green | green | green | green | green |
| Recruitment acceptable? | green | green | green | green | green | green | green | green | green | green | green | green | yellow |
| Exposure correctly measured? | green | green | green | green | green | green | green | green | green | green | green | green | yellow |
| Outcome correctly measured? | green | green | red | green | green | green | green | green | green | green | green | green | green |
| Identified confounding? | green | yellow | yellow | green | green | yellow | yellow | yellow | yellow | green | yellow | green | green |
| Addressed confounding? | green | red | yellow | green | green | green | green | green | green | green | green | green | green |
| Subject follow-up finish enough? | yellow | yellow | yellow | green | yellow | red | red | yellow | green | green | green | yellow | yellow |
| Subject follow-up long enough? | yellow | yellow | yellow | yellow | yellow | yellow | yellow | yellow | yellow | green | green | yellow | yellow |
| Results believeable? | green | green | green | green | green | green | green | green | green | green | green | green | green |
| Results comparable to others? | green | green | green | green | green | green | green | green | green | green | green | green | green |
| Implication for practice? | green | green | green | green | green | green | green | green | green | green | green | green | green |

| Yusheng Zhang, et al., 2020 [32] | Susan L Baumgartner, et al., 2021 [33] | N. Ramesh, et al., 2021 [34] | Li Jin-glong, et al., 2024 [16] | Gabriel P, et al., 2022 [35] | Ping Chen, et al., 2022 [36] | Lee, et al., 2023 [37] | Sana Maqbool, et al., 2023 [38] | Huan Wang, et al., 2023 [39] | Makan, H, et al., 2023 [40] | Siobhan Bourke, et al., 2023 [41] | Bridve Sivaku-mar, et al., 2023 [42] | Suguru Shimok-ihara, et al., 2023 [43] | Sarah Payne Riches, et al., 2024 [44] | Kazrin Ahmad, et al., 2024 [45] | Rahul Mittal, et al., 2024 [46] |
|---|---|---|---|---|---|---|---|---|---|---|---|---|---|---|---|

guideline, "all diabetics over the age of 60 are considered to have a high risk of suffering cardiovascular [16]."

Another common diabetic-related disease is hypo glycaemia, which is an overlooked medical problem that is popular among the diabetic elderly. It is reported that a high risk of serious hypo glycaemia episodes has a high connection with a long duration of type 2 diabetic elderly. Frequent hypo glycaemia will increase the possibility of frailty among the diabetic elderly [20]. A quantitative number of diabetic elderly who need to inject insulin can be frail, decline in vision or suffer from cognitive issues. Hence, diabetic elderly must arrange insulin dose injections depending on daily and hourly glucose fluctuations, which can benefit the hypo glycaemia prevention for diabetic elderly [20].

Additionally, as diabetic elderly age, type 2 diabetes also has an increasing impact on functional autonomy [29]. Long-term diabetes accelerates the loss of skeletal muscle mass and its function, both of which are essential to preventing frailty, sarcopenia, and incapacity from happening. While losing both can result in a decrease of mobility and quickness of movement. Hence, sarcopenia is regarded as a significant indicator to measure the development of limb impairment and frailty in diabetic elderly [26].

Finally, one of the major health issues for elderly individuals with diabetes is that prolonged diabetes can be linked to mild cognitive impairment and brain cortical abnormalities, leading to challenges in daily activities. This condition is associated with slowed mental, motor, and cognitive functions. Diabetes and dementia, including vascular dementia and Alzheimer's disease, are highly prevalent and often coexist in the elderly population [25].

**Table 4. Studies that focused on the health problems among diabetic elderly.**

| Author(s)/ Year | Thematic focus | Objectives | Country | Participants (Sample) | Data collection | Data analysis |
|---|---|---|---|---|---|---|
| A. Sinclair, T. Dunning, and L. Rodriguez-Mañas, 2015 [20] | Diabetic elderly's challenges | Examine the current state of knowledge about diabetes in older people and discuss how recognition of the effect of frailty and disability is beginning to lead to new management approaches. | UK | / | Literature review | Meta analysis |
| N. P. W. P. Sari and M. Manungkalit, 2019 [27] | Post prandial glucose predictor | Analyse the predictors of PPG level in diabetic elderly which consist of functional status, self-care activity, sleep quality, and stress level. | Indonesia | Old diabetes (N = 45) | Cross-sectional study | Pearson and Spearman Rank correlation test |
| A. M. Prina et al., 2019 [26] | Depression and Incidence of Frailty in Older People | Investigate the potential impact of depression on incident frailty in older people living in Latin America. | Latin America | Elderly diabetes (N = 12844) | Experiment | Statistical Analysis |
| I. Feinkohl et al., 2019 [25] | Metabolic syndrome and its components with cognitive impairment in older adults | Investigated the metabolic syndrome role in cognitive impairment. | Germany | Elderly diabetes (N = 202) | Experiment | Statistical Analysis |
| A. Pilotto, C. Custodero, S. Maggi, M. C. Polidori, N. Veronese, and L. Ferrucci, 2020 [29] | Frailty in older people | Summarize the current state of the art, the applications and the future directions of applicability of the CGA-based MPI for measuring frailty in older people. | Italy | Elderly diabetes (N = 192) | Literature review | Meta analysis |
| L. Jinglong, et al., 2024 [16] | Diabetic eldely healthy diet | Explores the potential application of health logos in the food service sector and its impact on consumer behaviors. | China | / | Literature review | Documentary |
| H. Wang et al., 2023 [39] | Diabetes care | Assess real-world cardiovascular safety for sulfonylureas, in comparison with dipeptidyl peptidase 4 inhibitors and thiazolidinediones. | UK | Elderly diabetes (N = 1100) | Experiment | Statistical Analysis |

## The application of IoT mobile app for diabetic elderly healthcare services

12 studies focused on the application of IoT mobile app for diabetic elderly healthcare services. The synthesis and details of these 12 studies are presented in Table 5, including publication year, thematic focus, research objectives, geographical context, methodological data particulars, and variables analyzed in each article.

Diabetes is a widespread chronic disease among the elderly population, presenting significant health challenges. The application of IoT mobile applications in healthcare offers promising solutions to enhance diabetes management for elderly patients. This section identified the findings on the application of IoT mobile apps in diabetic elderly healthcare services, then categorised the contributions into four fields: blood glucose monitoring, medication management, physical activity, and dietary monitoring.

**Table 5. Studies that focused on the application of IoT mobile app for diabetic elderly healthcare services.**

| Author(s)/ Year | Thematic focus | Objectives | Country | Participants (Sample) | Data collection | Data analysis |
|---|---|---|---|---|---|---|
| F. Mehravar, et al., 2016 [21] | Diabetes self-management and microvascular complications | Examine associations between diabetes self-management and microvascular complications in patients with type 2 diabetes. | Iran | 562 Iranian patients' elderly with type 2 diabetes | Cross-sectional study | Statistical analysis |
| Karandeep, et al., 2016 [22] | Mobile App | Patient-Facing Mobile Apps to Treat High-Need, High-Cost Populations: A Scoping Review | US | / | Literature review | Scoping review |
| E. Toschi and M. N. Munshi, 2020 [30] | Benefit of technology in diabetic elderly | Understand the benefits and barriers in the use of technology in the aging population | / | / | Literature review | Documentary |
| Y. Zhang et al., 2020 [32] | Self-monitoring of blood glucose | Investigated the association between glucose variation indices estimated by SMBG and b-cell function among Chinese patients with type 2 diabetes mellitus (T2DM). | China | 397 patients with type 2 diabetes | Cross sectional study | Statistical analysis |
| Arriel Benis et al., 2020 [31] | Digital communications | Communication Behavior Changes Between Patients With Diabetes and Healthcare Providers Over 9 Years:Retrospective Cohort Study | Israel | 168 patients diagnosed with diabetes | Retrospective cohort study | Statistical analysis |
| S. L. Baumgartner, et al., 2021 [33] | Novel Digital Pill System | Evaluate a novel digital pill system, the ID-Cap System from etectRx, for usability among patient users in a simulated real-world use environment. | US | 17 patients with diverse backgrounds | Experiment | SPSS |
| P. Chen, et al., 2022 [36] | Wearable tracker | Explore the predicting factors of Chinese elderly type 2 diabetic patients' adoption intention to wearable activity trackers and their actual wearing behavior. | US | 725 diabetic patients | Survey | Structural equation modeling |
| Makan, H, et al., 2023 [40] | MyDiaCare, Digital Tools | Clinical and Economic Assessment of MyDiaCare, Digital Tools Combined With Diabetes Nurse Educator Support, for Managing Diabetes in South Africa: Observational Multicenter, Retrospective Study Associated With a Budget Impact Model. | South Africa | 117 patients | observational, retrospective, multicenter, single-group study | Statistical analysis |
| S. Maqbool, et al., 2023 [38] | Sensing Technologies for diabetic elderly | Smart sensing technologies-based architecture is proposed that uses AI and the Internet of Things (IoT) for continuous monitoring and health assistance for diabetes patients. | Pakistan | Elderly diabetes using technology (N = 50) | Experiment | Statistical analysis |
| B. Sivakumar, et al., 2023 [42] | Mobile app | Determine patients' needs, motivations, and challenges on the use ofmobile apps tosupport HF management. | Canada | Elders (N = 19) | Interview | Atlas.ti |
| J. J. Lee, et al., 2023 [37] | M-Health apps | Assess the content and quality offree, popular mHealth apps supporting plant-based diets for Canadians. | Toronto | 16 popular, freely available plant-based diet apps | Cross-sectional content | Quality analysis |
| R. Mittal, et al., 2024 [46] | Blood glucose monitoring devices | Assume greater control over their health, alleviating the burdens associated with their condition. | US | / | Literature review | Documentary |

Blood glucose monitoring is a critical component of diabetes management, especially for elderly patients [32]. IoT-enabled devices, such as smart glucometers and continuous glucose monitors (CGMs), facilitate real-time tracking and management of blood glucose levels. These devices enable continuous monitoring of glucose levels, alerting patients and healthcare providers to any abnormal readings, which is crucial for preventing severe hypoglycemia and hyperglycemia. Data collected by these devices can be integrated with mobile applications for comprehensive analysis and trend identification. This integration aids in personalized treatment adjustments and better diabetes management [46]. Additionally, IoT devices can send real-time alerts and notifications to patients and caregivers, ensuring timely interventions and reducing the risk of complications. Studies have shown that using IoT-enabled glucose monitoring devices significantly improves glycemic control among elderly patients [38]. For instance, a study by Toschi and Munshi [30] found that elderly patients using CGMs experienced fewer episodes of severe hypoglycemia compared to those using traditional monitoring methods. The use of IoT-enabled mobile applications for diabetic elderly healthcare services has been explored as a promising solution to improve care outcomes and enable more effective monitoring. Another prominent example comes from Makan's research, who present the case of MyDiaCare, a mobile application specifically designed for diabetes management in South Africa. The app combines real-time glucose monitoring with automatic alerts sent to caregivers and healthcare professionals, allowing for timely interventions. The study found that this approach significantly reduced hospital admissions for elderly patients by facilitating better glycemic control, even in remote areas with limited healthcare access [40].

Medication adherence is another critical aspect of diabetes management for elderly patients. IoT applications for medication management offer various features to enhance adherence and ensure timely medication intake. These applications can send reminders to patients to take their medications as prescribed, reducing the likelihood of missed doses [30]. Smart pill dispensers and connected apps track medication intake, providing data on adherence patterns that can be shared with healthcare providers for better management [42]. IoT apps can also notify caregivers if a patient misses a dose, allowing for prompt intervention and support. Research indicates that IoT-enabled medication management tools improve adherence rates among elderly diabetic patients. A study by Baumgartner et al [33] found that elderly patients using smart pill dispensers had a 20% higher medication adherence rate compared to those using traditional methods. The integration of mobile apps into routine care has also been shown to improve patient self-management. For instance, Benis evaluated how the communication behavior between patients and healthcare providers changed when IoT mobile apps were introduced. The study indicated that elderly patients with access to mobile health apps experienced improved communication with their healthcare team, resulting in more personalized care plans and a higher rate of medication adherence. This was particularly effective in cases where patients were provided with simplified interfaces and training to overcome the initial technological barriers [31].

Maintaining a healthy lifestyle through regular physical activity is essential for diabetes management. IoT devices such as fitness trackers play a crucial role in monitoring and promoting physical activity. Fitness trackers monitor physical activities, providing data on steps taken, calories burned, and active minutes. This data helps patients and healthcare providers assess and adjust activity levels to improve diabetes management [21]. Studies support the efficacy of IoT applications in promoting physical activity among elderly diabetic patients. A study by Chen et al [36] found that elderly patients using fitness trackers increased their physical activity levels by 30%. A notable empirical example is the collaboration between health services and tech companies to deploy mobile apps in large-scale trials. Karandeep illustrate how a pilot program in the UK incorporated IoT mobile apps to track health metrics

of diabetic elderly patients. The study showed that after six months of continuous use, patients reported improved quality of life, and healthcare providers noted a significant reduction in emergency visits related to diabetic complications. This case underlines the practical applicability of IoT mobile apps in improving the quality of life for diabetic elderly patients, while also reducing healthcare costs by preventing complications [22].

A balanced diet is crucial for diabetes management, and IoT devices like smart scales and dietary monitoring apps assist in this area. Mobile applications track food intake, provide nutritional information, and offer personalized dietary recommendations, helping elderly patients maintain a balanced diet and control blood glucose levels. By integrating data from various sources, IoT applications provide a comprehensive view of a patient's health, facilitating more effective management strategies. Studies demonstrate that dietary monitoring apps help patients reduce their HbA1c levels by providing personalized feedback and support [37]. Makan et al. detail how the Diabetic App was deployed in a South African pilot study targeting elderly diabetic patients. The app integrates with wearable devices to monitor physical activity and blood sugar levels, providing real-time insights into the impact of dietary choices. For example, after meals, the app offers feedback on whether the user's blood glucose level has stayed within an optimal range, helping patients to make immediate adjustments to their diet. Over a six-month period, users of the Diabetic App showed significant improvement in their blood glucose control and reported higher adherence to a balanced diet plan. The study highlighted a 15% reduction in HbA1c levels, reflecting better long-term glucose management. The Diabetic App also includes educational features that guide patients in making healthier food choices. By using machine learning algorithms, the app predicts the potential impact of various meals on a patient's blood sugar, providing alternative meal suggestions to reduce spikes in glucose. This balanced diet management system, supported by real-time feedback, made a substantial difference in the quality of life of elderly diabetic patients, many of whom faced challenges in adhering to strict dietary plans prior to using the app [40]. This example demonstrates the practical applicability of IoT-based mobile apps in facilitating better diet management for diabetic elderly patients, contributing to improved health outcomes and long-term disease management.

In conclusion, the successful implementation of IoT mobile applications has significantly improved diabetic care for elderly patients across multiple areas, including blood glucose monitoring, medication management, physical activity, and dietary monitoring. These technologies enable real-time tracking, personalized feedback, and timely interventions, which enhance self-management and communication with healthcare providers. Studies consistently show improved health outcomes, including better glycemic control, higher medication adherence, increased physical activity, and better diet management, all of which contribute to a higher quality of life and reduced complications for elderly diabetic patients.

## The challenges of IoT mobile app for diabetic elderly healthcare services

10 studies focused on the challenges of IoT mobile app for diabetic elderly healthcare services. The synthesis and details of these 10 studies are presented in Table 6, including publication year, thematic focus, research objectives, geographical context, methodological data particulars, and variables analyzed in each article.

Diabetes, as a pervasive chronic disease among the elderly population, demands continuous and meticulous management, which can be particularly difficult for senior patients. While IoT mobile applications provide innovative solutions, significant challenges still impede their effective use and integration into elderly healthcare services. This section delves into the most critical challenges identified in the literature, addressing both technical and non-technical

barriers, and explores potential solutions for overcoming these issues. This part investigated the current challenges and barriers to the application of IoT mobile apps in diabetic elderly healthcare services, then classified them into four fields: interoperability, data security and privacy, usability, and involvement of policymakers.

Interoperability remains a critical barrier in the deployment of IoT applications for diabetic care. Devices from different manufacturers often use incompatible communication protocols, leading to difficulties in integrating data across platforms. This lack of standardization hinders the seamless sharing of health data between healthcare providers, apps, and patients [24]. For instance, Ramesh emphasized the need for uniform communication standards to ensure that patient data such as glucose levels and medication information can be accurately transmitted and interpreted across different devices and platforms. Without interoperability, the risk of incomplete or inaccurate data increases, which could compromise patient safety and the quality of care provided [34]. To address this, future research could focus on developing frameworks that standardize data formats and communication protocols, ensuring compatibility between various IoT devices. Additionally, policy interventions led by healthcare regulators could mandate specific standards for interoperability, thus reducing the risk of fragmented healthcare data. Healthcare professionals and device manufacturers should collaborate to develop unified systems that prioritize patient safety and data accuracy, a step critical for long-term IoT integration [35].

Data security and privacy stand as central concerns in the adoption of IoT mobile apps for elderly diabetic care. These devices gather sensitive health information such as

**Table 6. Studies that focused on the challenges of IoT mobile app for diabetic elderly healthcare services.**

| Author(s)/ Year | Thematic focus | Objectives | Country | Participants (Sample) | Data collection | Data analysis |
|---|---|---|---|---|---|---|
| Amine, et al., 2014 [19] | Insulin therapy for diabetes | Develop a methodology of data processing for following the insulin therapy at home. | France | 71 patients | Experiment | Statistical analysis |
| Mousumi Sircar, et al., 2016 [23] | Hypoglycemia in the Older Adult | Review of Hypoglycemia in the Older Adult | US | / | Literature review | Documentary |
| T. Alanzi, 2018 [24] | M-Health for diabetes self-management | Identify the barriers to mHealth for diabetes care in the Kingdom and the relevant solutions are discussed. | Saudi Arabia | Older Adults With diabetes (N = 40) | Survey | Statistical analysis |
| Li J, et al., 2020 [28] | A Mobile-Based Intervention for Glycemic Control | Evaluate the effectiveness of a mobile-based intervention on glycemic control in patients with type 2 diabetes based on real-world population data. | China | 2400 patients | Experiment | Statistical analysis |
| Payne Riches S, et al., 2021 [44] | A Mobile Health intervention | Assess the feasibility of a complex behavioral intervention to lower salt intake in people with elevated blood pressure and test the trial procedures for a randomized controlled trial to investigate the intervention's effectiveness. | UK | 47 participants | Experiment | Statistical analysis |
| Ramesh N, et al., 2021 [34] | Rule-Based Recommender System | A novel rule-based model is proposed for recommending foods for Indian elderly diabetic population based on Glycemic Index (GI) of food items. | Indian | 54 diabetic elderly | Questionnaire | Statistical analysis |
| Pinheiro GPM, et al., 2022 [35] | Wearable Health Device | proposes a complete framework for health systems composed of multi-sensor wearable health devices (MWHD), high-resolution parameter estimation, and real-time monitoring applications. | Germany | Duabetic elderly | Experiment | Statistical analysis |
| Bourke S, et al., 2023 [41] | Diabetes care | This qualitative study explored the current barriers and enablers ofdiabetes care inthe Indian Ocean Territories (IOT). | Australia | 20 participants | Interview | Qualitiative analysis |
| Shimokihara et al., 2023 [43] | Mobile application | Explore clinical factors associated with navigation walking in older adults | Japan | 20 community-dwelling older adults | Experiment | Statistical analysis |
| Ahmad et al., 2024 [45] | Internet of things | To identify the dominant barriers of IoT implementation. | Bangladesh | / | Literature review | Documentary |

blood glucose readings, medication schedules, and lifestyle habits, making them prime targets for potential data breaches. Li et al. highlighted that elderly patients often lack the technical knowledge to assess the security risks associated with IoT apps [28]. Moreover, many patients expressed concern about who has access to their data and how it is stored, particularly in cases where app developers might not fully comply with healthcare data regulations [23]. Addressing these concerns requires more than technical fixes like encryption; it involves educating elderly users on the risks and benefits of IoT health services. Benis Bourke suggested that one way to increase user confidence is by developing clear, easy-to-understand consent protocols that inform patients of how their data is used and protected. Furthermore, introducing regulatory frameworks that ensure strict compliance with data protection standards can build trust among elderly users and their caregivers, making them more likely to adopt these technologies. Future research could explore the development of decentralized systems, such as blockchain technology, to further enhance data security and user transparency [41].

Usability issues also pose a major barrier to the successful deployment of IoT mobile apps among elderly patients. The elderly population often struggles with complex interfaces and lacks the technical literacy needed to navigate advanced applications [44]. Shimokihara reported that 30% of the elderly participants in their study required assistance to use the MyDiaCare app. This reflects a broader issue where elderly patients find it difficult to engage with apps that are not tailored to their cognitive and physical needs [43]. To overcome this challenge, developers need to co-design mobile apps with elderly users in mind, ensuring that the interface is simplified, intuitive, and responsive to their limitations. For example, incorporating voice commands or larger, easily readable fonts could help address these usability barriers. Moreover, healthcare professionals play a key role in facilitating user adoption by offering training sessions and ongoing support to elderly patients. Personalized onboarding experiences that familiarize patients with the app's functions can significantly improve user adoption and engagement over time [19].

Healthcare professionals and policymakers play crucial roles in overcoming the challenges of IoT adoption. Benis et al. emphasized that healthcare providers should be trained to guide elderly patients through the use of mobile health applications, ensuring they feel comfortable with the technology and understand its benefits. Additionally, policymakers need to implement guidelines that promote the standardization of IoT devices, enforce data privacy regulations, and ensure that elderly patients are adequately supported in their use of mobile healthcare apps [45].

In conclusion, while IoT mobile applications offer promising solutions for diabetic care in the elderly, significant challenges such as interoperability, data security, usability, and policy involvement remain. Future research should delve deeper into developing standardized frameworks for device compatibility and data integration, as well as explore advanced security measures, such as blockchain, to enhance data privacy. Addressing usability issues requires co-designing apps with elderly users to improve accessibility and support. Additionally, involving healthcare professionals and policymakers in promoting training, standardization, and regulatory oversight will be critical for the successful and widespread adoption of these technologies.

## Discussion

IoT mobile applications hold significant promise for enhancing diabetes management in elderly patients through improved monitoring, medication management, and lifestyle support. These technologies offer innovative solutions that can facilitate real-time tracking of blood glucose levels, ensure timely medication adherence, and promote healthy lifestyle

choices. The potential benefits include better glycemic control, reduced complications, and improved overall quality of life for diabetic elderly individuals.

However, several challenges must be systematically addressed to fully realize the potential of IoT mobile applications in diabetic elderly healthcare services. Technical issues such as interoperability, data security, and reliability remain significant barriers. To overcome these challenges, future research should focus on developing standardized frameworks for device compatibility, ensuring seamless communication between different devices and platforms. In terms of data security, regulatory compliance should be prioritized to protect sensitive health information, with solutions such as blockchain or decentralized systems offering potential ways to enhance data protection and transparency. Additionally, maintaining consistent connectivity and device accuracy is essential for effective diabetes management.

User adoption and usability also present considerable challenges. Elderly patients often face cognitive and physical limitations that can hinder their ability to effectively use IoT technologies [47]. Future solutions must prioritize designing user-friendly interfaces with simplified device operations, larger text, and voice commands [48]. Moreover, providing adequate training and support for elderly patients, as well as healthcare professionals guiding them, is essential to ensure long-term engagement and motivation. Personalized onboarding experiences, as well as ongoing support, can improve adoption rates and usage [49].

Furthermore, managing and analyzing the vast amounts of data generated by IoT devices is a complex task. Healthcare providers need robust systems to store, manage, and analyze this data to provide actionable insights and personalized care. Future research should explore frameworks that ensure both technological advancements and user-centered design are prioritized in the development and deployment of IoT solutions for elderly care. Policymakers and healthcare professionals must play an active role in ensuring that these systems are not only compliant with privacy regulations but also designed to enhance patient autonomy and care outcomes.

## Conclusion

This paper comprehensively investigated the current development and application of IoT mobile technologies in healthcare services for diabetic elderly patients, highlighting both the benefits and the challenges. The findings demonstrate that IoT mobile applications have the potential to transform diabetes management for elderly individuals by improving monitoring, medication adherence, and lifestyle management. However, addressing the challenges related to interoperability, data security, usability and involvement of policymakers is crucial for effective implementation. To fully harness the potential of IoT technologies in diabetic care, collaborative efforts from device manufacturers, healthcare providers, and policymakers are essential. Developing standardized communication protocols to ensure seamless integration across platforms, implementing robust data protection measures, and designing user-friendly interfaces tailored to elderly patients will mitigate the technical barriers. Providing comprehensive training and support for elderly patients and caregivers will further enhance user adoption and sustained engagement. However, the findings of this systematic review were limited by the inclusion of only 29 articles, which may restrict the generalizability of the conclusions. Future research and continued development in these areas will ensure that IoT mobile applications can significantly improve the health and quality of life for diabetic elderly patients as well as generalizability.

## Supporting information

**S1 File. Includes: Section 1: PRISMA checklist, Section 2: Detailed inclusion/exclusion criteria, Section 3: Literature search syntax.**
(ZIP)

**S2 File. Includes: Sheet: 4: Table of studies identified and excluded from analysis, 5: Data extraction table from primary sources (Sheet: CASP) includes the reporting quality assessment, 6: (Sheet: Cochrance RoB1) includes the risk of bias assessment, and 7: (Sheet: Included & excluded studies) includes full-text screened studies with the reason(s) of exclusion.**
(ZIP)

## Author contributions

**Conceptualization:** Faisul Arif Ahmad, Qisen Zhu.

**Formal analysis:** Rosalam Che Me.

**Funding acquisition:** Rosalam Che Me.

**Investigation:** Rosalam Che Me, Qisen Zhu.

**Software:** Qisen Zhu.

**Supervision:** Faisul Arif Ahmad.

**Writing – original draft:** Jinglong Li.

**Writing – review & editing:** Jinglong Li, Rosalam Che Me, Faisul Arif Ahmad.

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
