## [Decision Letter · Decision Letter 0]

8 Sep 2024

PONE-D-24-34100Investigating the Application of IoT Mobile App and Healthcare Services for Diabetic Elderly: A Systematic ReviewPLOS ONE

Dear Dr. Li,

Thank you for submitting your manuscript to PLOS ONE. After careful consideration, we feel that it has merit but does not fully meet PLOS ONE’s publication criteria as it currently stands. Therefore, we invite you to submit a revised version of the manuscript that addresses the points raised during the review process.

We look forward to receiving your revised manuscript.

Kind regards,

Muhammad Zulkifl Hasan, PhD

Academic Editor

PLOS ONE

 [This research was funded by a grant from Universiti Putra Malaysia Geran Putra grant (GP-IPS). NO: [GP-IPS/2023/9772400].].  

3. As required by our policy on Data Availability, please ensure your manuscript or supplementary information includes the following: 

Additional Editor Comments (if provided):

Reviewers' comments:

Reviewer's Responses to Questions

**Comments to the Author**

1. Is the manuscript technically sound, and do the data support the conclusions?

Reviewer #1: Yes

Reviewer #2: Yes

2. Has the statistical analysis been performed appropriately and rigorously? 

Reviewer #1: Yes

Reviewer #2: Yes

3. Have the authors made all data underlying the findings in their manuscript fully available?

Reviewer #1: Yes

Reviewer #2: Yes

4. Is the manuscript presented in an intelligible fashion and written in standard English?

Reviewer #1: Yes

Reviewer #2: Yes

5. Review Comments to the Author

Reviewer #1: The manuscript titled "Investigating the Application of IoT Mobile App and Healthcare Services for Diabetic Elderly: A Systematic Review" effectively addresses a critical health issue but has some notable weaknesses. Firstly, the scope of the study is somewhat limited by the relatively small sample size of 18 articles reviewed. This constraint restricts the generalizability of the findings, as the study only captures a narrow snapshot of the available literature on IoT mobile applications in healthcare for the diabetic elderly. Moreover, the study primarily focuses on positive outcomes, while not giving sufficient attention to more granular aspects of the challenges, particularly those beyond technical interoperability and data security issues.

Secondly, the methodology employed for selecting and reviewing the articles could have been more rigorous. Although the paper mentions the use of CASP's Quality Appraisal Tool, it does not provide detailed insights into how each study was critically evaluated. This omission weakens the reliability of the quality assessment, leaving readers with limited understanding of the varying quality of the studies included. Additionally, while the discussion highlights key challenges such as data privacy and usability issues for elderly patients, it lacks depth in exploring how these challenges can be systematically addressed through future research and implementation.

Reviewer #2: The manuscript "Investigating the Application of IoT Mobile App and Healthcare Services for Diabetic Elderly: A Systematic Review" offers valuable insights into the use of IoT mobile applications for managing diabetes in elderly populations. However, the paper exhibits several weaknesses that need to be addressed to improve its clarity and impact. First, while the authors highlight the potential of IoT technologies, the paper lacks depth in terms of real-world case studies or concrete examples demonstrating the successful implementation of these technologies in diabetic care. The lack of empirical data makes it difficult to evaluate the practical applicability of the findings. Adding specific case studies or implementation results would strengthen the claims and provide a more well-rounded discussion.

Additionally, the review touches on various challenges such as interoperability, user adoption, and data security, but these issues are discussed superficially. There is little detailed exploration of how these challenges can be addressed, or the role of healthcare professionals and policymakers in overcoming them. To improve the manuscript, the authors should delve deeper into potential solutions or frameworks for tackling these barriers, such as designing user-friendly interfaces for elderly patients or ensuring regulatory compliance for data protection. A more structured approach to discussing these challenges would offer greater practical value and make the findings more actionable for stakeholders in the healthcare industry.

Lastly, the selection criteria for articles in the systematic review are not clearly justified. While the paper provides some information about the databases used and the inclusion criteria, it would benefit from a more rigorous explanation of why certain studies were selected and how their findings were synthesized. The lack of a critical appraisal of the studies weakens the review's overall credibility. Incorporating a stronger methodology section that outlines the study quality and relevance of the included articles would enhance the reliability of the conclusions drawn in the manuscript.

6. PLOS authors have the option to publish the peer review history of their article (what does this mean? ). If published, this will include your full peer review and any attached files.

**Do you want your identity to be public for this peer review?** For information about this choice, including consent withdrawal, please see our Privacy Policy .

Reviewer #1: **Yes: ** Muhammad Zunnurain Hussain

Reviewer #2: No

---

## [Author Response · Author response to Decision Letter 1]

14 Oct 2024

I am grateful for your insightful comments. I have been able to incorporate changes to reflect all of the suggestions provided by you before. I have also highlighted the changes in the revised manuscript with track changes for your review.

Deatiled respond to reviewers document can be downloaded in supporting information [RESPONSE TO REVIEWERS ], thanks a lot of your comments on improving my manuscript.

---

## [Decision Letter · Decision Letter 1]

17 Jan 2025

PONE-D-24-34100R1Investigating the Application of IoT Mobile App and Healthcare Services for Diabetic Elderly: A Systematic ReviewPLOS ONE

Dear Dr. Li,

Thank you for submitting your manuscript to PLOS ONE. After careful consideration, we feel that it has merit but does not fully meet PLOS ONE’s publication criteria as it currently stands. Therefore, we invite you to submit a revised version of the manuscript that addresses the points raised during the review process. Authors are instructed to look for the queries raised by the reviewer for the further round of the consideration.

We look forward to receiving your revised manuscript.

Kind regards,

Vishal Sorathiya

Academic Editor

PLOS ONE

Journal Requirements:

Comments from PLOS Editorial Office: We note that one or more reviewers has recommended that you cite specific previously published works. As always, we recommend that you please review and evaluate the requested works to determine whether they are relevant and should be cited. It is not a requirement to cite these works. We appreciate your attention to this request.

Reviewers' comments:

Reviewer's Responses to Questions

**Comments to the Author**

1. If the authors have adequately addressed your comments raised in a previous round of review and you feel that this manuscript is now acceptable for publication, you may indicate that here to bypass the “Comments to the Author” section, enter your conflict of interest statement in the “Confidential to Editor” section, and submit your "Accept" recommendation.

Reviewer #1: All comments have been addressed

Reviewer #3: (No Response)

2. Is the manuscript technically sound, and do the data support the conclusions?

Reviewer #1: Yes

Reviewer #3: Yes

3. Has the statistical analysis been performed appropriately and rigorously? 

Reviewer #1: Yes

Reviewer #3: N/A

4. Have the authors made all data underlying the findings in their manuscript fully available?

Reviewer #1: Yes

Reviewer #3: Yes

5. Is the manuscript presented in an intelligible fashion and written in standard English?

Reviewer #1: Yes

Reviewer #3: Yes

6. Review Comments to the Author

Reviewer #1: Thank you for revising your manuscript and addressing the previous feedback. After carefully reviewing the changes, I am pleased to acknowledge that most of the concerns raised have been effectively addressed.

Reviewer #3: 1. Introductions need to improve with recent work on Diabetes/IOT/Diabetes in Elders like; https://doi.org/10.1111/hsc.13522,
https://doi.org/10.3389/fpubh.2022.871575,
http://dx.doi.org/10.1109/TR.2023.3336330,
https://doi.org/10.1109/TIT.2023.3340147

2. The table 2 need to cite, form where or what based this inclusion and exclusion selected.

3. Table 3,4,5, 6; each raw need to cite as per journal style [..]

4. . Li highlighted that elderly patients often lack the technical knowledge to assess the security risks

associated with IoT apps (35). Here Li et. al. should be there in place of Li. Please check the manuscript for such error.

5. User adoption and usability also present considerable challenges. Elderly patients often face

cognitive and physical limitations that can hinder their ability to effectively use IoT technologies.

Future solutions must prioritize designing user-friendly interfaces with simplified device operations,

larger text, and voice commands. Moreover, providing adequate training and support for elderly

patients, as well as healthcare professionals guiding them, is essential to ensure long-term engagement

and motivation. Personalized onboarding experiences, as well as ongoing support, can improve

adoption rates and usage. This paragraph needs to cite, with appropriate support reference.

7. PLOS authors have the option to publish the peer review history of their article (what does this mean? ). If published, this will include your full peer review and any attached files.

**Do you want your identity to be public for this peer review?** For information about this choice, including consent withdrawal, please see our Privacy Policy .

Reviewer #1: **Yes: ** Muhammad Zunnurain Hussain

Reviewer #3: No

---

## [Author Response · Author response to Decision Letter 2]

18 Jan 2025

RESPONSE TO REVIEWERS – Minor Revision

1. Reviewer #3: Introductions need to improve with recent work on Diabetes/IOT/Diabetes in Elders like; https://doi.org/10.1111/hsc.13522,
https://doi.org/10.3389/fpubh.2022.871575,
http://dx.doi.org/10.1109/TR.2023.3336330,
https://doi.org/10.1109/TIT.2023.3340147

RESPONSE: Thank you for your valuable feedback. I have revised the introduction to incorporate recent work on diabetes, IoT, and diabetes in elderly populations, as per your suggestion. Specifically, I have added the following 4 comprensive citations recommended to strengthen the introduction, referring to the pages [3,4] which revised. Thank you again for your insightful comments.

2. Reviewer #3: The table 2 need to cite, form where or what based this inclusion and exclusion selected.

RESPONSE:Thank you for your advice, I added a citation in Table 2 to show the source reference, you can see the revision in page 5-Table 2.

3.Reviewer #3: Table 3,4,5, 6; each raw need to cite as per journal style [..]

RESPONSE: Thank you for your suggestin, I revised and added each row of per journal citation in Table 3,4,5, 6; you can see the changes in page 7 (Table 3), page 9 (Table 4), page 11 (Table 5) and page 15 (Table 6), where highlighted in red.

4. Reviewer #3: Li highlighted that elderly patients often lack the technical knowledge to assess the security risks

associated with IoT apps (35). Here Li et al. should be there in place of Li. Please check the manuscript for such error.

RESPONSE: Thank you for your careful suggestion, I revised based on your advice, You can find the revision in the updated manuscript on pages 17-line 10.

5. Reviewer #3: [User adoption and usability also present considerable challenges. Elderly ...... rates and usage.] This paragraph needs to cite, with appropriate support reference.

RESPONSE: Thank you for your important suggestion, I revised and added related citation to this paragraph to support my iead. You can find the revision with citation in the updated manuscript on pages 19-line 18, 20, 23.

---

## [Decision Letter · Decision Letter 2]

2 Mar 2025

Investigating the Application of IoT Mobile App and Healthcare Services for Diabetic Elderly: A Systematic Review

PONE-D-24-34100R2

Dear Dr. Li,

We’re pleased to inform you that your manuscript has been judged scientifically suitable for publication and will be formally accepted for publication once it meets all outstanding technical requirements.

Kind regards,

Vishal Sorathiya

Academic Editor

PLOS ONE

Additional Editor Comments (optional):

Reviewers' comments:

Reviewer's Responses to Questions

**Comments to the Author**

1. If the authors have adequately addressed your comments raised in a previous round of review and you feel that this manuscript is now acceptable for publication, you may indicate that here to bypass the “Comments to the Author” section, enter your conflict of interest statement in the “Confidential to Editor” section, and submit your "Accept" recommendation.

Reviewer #3: (No Response)

2. Is the manuscript technically sound, and do the data support the conclusions?

Reviewer #3: Yes

3. Has the statistical analysis been performed appropriately and rigorously? 

Reviewer #3: Yes

4. Have the authors made all data underlying the findings in their manuscript fully available?

Reviewer #3: Yes

5. Is the manuscript presented in an intelligible fashion and written in standard English?

Reviewer #3: Yes

6. Review Comments to the Author

Reviewer #3: (No Response)

7. PLOS authors have the option to publish the peer review history of their article (what does this mean? ). If published, this will include your full peer review and any attached files.

**Do you want your identity to be public for this peer review?** For information about this choice, including consent withdrawal, please see our Privacy Policy .

Reviewer #3: No

---

## [Editor Report · Acceptance letter]

PONE-D-24-34100R2

PLOS ONE

Dear Dr. Li,

I'm pleased to inform you that your manuscript has been deemed suitable for publication in PLOS ONE. Congratulations! Your manuscript is now being handed over to our production team.

Kind regards,

on behalf of

Dr. Vishal Sorathiya

Academic Editor

PLOS ONE